# Black Men and Health Literacy: Strategies for Improvement in a Digital Age Through the Adaptation of a Chronic Disease Self-Management Program

**DOI:** 10.3390/ijerph22071153

**Published:** 2025-07-21

**Authors:** Evelina Weidman Sterling, Laura Stevens, Vanessa Robinson-Dooley, Tyler Collette

**Affiliations:** 1Radow College of Humanities and Social Sciences, Department of Sociology and Criminal Justice, Kennesaw State University, Marietta, GA 30060, USA; 2School of Social Work, Simmons University, Boston, MA 02115, USA; laura.stevens@simmons.edu (L.S.); vanessa.robinson-dooley@simmons.edu (V.R.-D.); 3Radow College of Humanities and Social Sciences, Department of Psychological Science, Kennesaw State University, Marietta, GA 30060, USA; tcollet1@kennesaw.edu

**Keywords:** Black men, minority health, health literacy, chronic disease self-management, digital health, health communication, health equity, social determinants of health

## Abstract

Health literacy is a critical determinant of health outcomes, yet it is often overlooked, particularly among marginalized groups. This paper explores the significance of health literacy, with a particular focus on low-income African American and Black (AA/B) men, a population that faces unique challenges due to intersecting factors such as race, gender, socioeconomic status, and educational disparities. We examine how these factors contribute to health literacy gaps, highlighting adverse effects on health outcomes for AA/B men compared to the general population. Additionally, we stress the growing importance of digital literacy in an increasingly technology-driven world. Not actively addressing digital health literacy, especially within chronic disease self-management programs (CDSMPs), further exacerbates health disparities within this group. Recommendations are provided for improving health literacy, with specific strategies to also enhance general literacy and digital literacy, among low-income AA/B men. The paper also advocates for a systematic review of the existing literature on health literacy among this group, emphasizing the need for tailored interventions that account for the unique challenges faced by low-income AA/B men. In conclusion, the paper underscores the critical need for targeted research and practical approaches to improve health literacy and ultimately health outcomes for AA/B men in the digital age, particularly through CDSMPs.

## 1. Introduction

Health literacy aids individuals in promoting and attaining good health and is vital to achieving health equity [1,2]. An estimated 87 million adults have low health literacy [3]. Low health literacy has been associated with negative health outcomes such as longer hospital stays [4,5,6], problems with medication adherence, poor self-care, greater post-surgical complications, and higher mortality rates [5,7,8]. Low health literacy has also been linked to poor health outcomes and health disparities [7,9,10]. In addition to negative health outcomes, the economic consequences of poor health literacy are also great, in that poor health literacy is projected to cost USD 1.6 to 3.6 million [3]. Consequently, the goal of this research is to investigate health literacy as a critical component of successfully adapting a chronic disease self-management program (CDSMP) specifically for low-income African American and Black (AA/B) men living with multiple chronic conditions, including assessing feasibility, acceptability, and preliminary impact.

### 1.1. Defining Health Literacy

While the term “literacy” means the ability to read and write, the term health literacy includes much more. “Health literacy” first appeared in 1974 in the article “Health Education as Social Policy” by Dr. Scott K. Simonds, where he argued that students exposed to health education in school can achieve literacy in health [11,12,13]. Since then, the concept of health literacy has evolved [12,13,14,15]. Still, inconsistency in the definition exists [15]. In 2010, the Healthy People Initiative run by the U.S. Department of Health and Human Services’ (HHS) Office of Disease Prevention and Health Promotion (ODPHP) sought to improve health literacy throughout the United States [16]. This led to a wider definition of health literacy that describes an individual’s ability to obtain, process, and understand health information to make appropriate health decisions [17].

Although definitions of health literacy vary, most definitions center on a person’s skills or abilities to obtain or understand health information to make appropriate healthcare decisions promoting and improving health [12,17,18]. One of the most widely accepted definitions of health literacy includes “The ability of an individual to obtain and translate knowledge and information to maintain and improve health in a way that is appropriate to the individual and system contexts” [19].

The Centers for Disease Control and Prevention (CDC) encourages both personal health literacy and organizational health literacy. Personal health literacy is “the degree to which individuals have the ability to find, understand, and use information and services to inform health-related decisions and actions for themselves and others” [2]. Organizational health literacy is “the degree to which organizations equitably enable individuals to find, understand, and use information and services to inform health-related decisions and actions for themselves and others” [2]. The CDC definition is an update from prior definitions that only centered on individual understanding of health information to one that includes the individual’s application of healthcare information and the ability of individuals to make well-informed decisions. Additionally, the CDC definition calls upon organizations to address health literacy as an issue of equity.

### 1.2. Understanding Health Literacy for Marginalized Groups

Low health literacy affects vulnerable and marginalized populations at greater rates [20,21,22]. A 2006 report from the U.S. Department of Education found that adults living below the poverty level; Black, Hispanic, American Indian/Alaska Native, and multiracial adults; non-English speakers; older adults; and men had lower levels of health literacy [20]. In the 2003 National Assessment of Adult Literacy, 58% of African Americans demonstrated “below basic or basic health literacy” compared to 28% of Whites and 31% of Asian/Pacific Islanders [20]. Health literacy rates were especially low among AA/B men [20]. Systemic issues such as racism, medical mistrust, inequity in healthcare access, and a lack of educational opportunities may impact health literacy for AA/B men [23,24,25,26,27,28]. Although unique inequities exist for AA/B men, this problem often goes unrecognized by medical professionals [29].

For AA/B men, health literacy may influence other factors such as quality of life and physical and mental well-being. Low health literacy was found to be more common among minorities [20,30], exacerbating health disparities for AA/B men [31,32]. In the general population, limited health literacy has been associated with negative health outcomes, such as longer hospital stays [4,5,6], poor self-care, greater post-surgical complications, and higher mortality rates [5,7,8]. Similarly, African Americans’ experiences with lower levels of health literacy have led to lower medication adherence [33,34]. Moreover, low health literacy was associated with a lower quality of life for minority men [34]. Alternatively, better physical and mental well-being is correlated with higher health literacy [30]. These findings emphasize the need to improve health literacy for AA/B men who are at greater risk for chronic health conditions. Health literacy may serve as a protective factor in reducing racial health disparities for chronic conditions, both physical and mental [33].

Health literacy may also influence actual health beliefs and behaviors among AA/B men, such as ignoring life-saving preventative care like colorectal cancer (CRC) screening due to strong fatalistic beliefs related to cancer. For AA/B men, cultural variables can influence provider trust, feelings of hopelessness, health temporal orientation, health literacy, and medical care intentions [35]. In addition, research shows that health literacy is negatively associated with fatalism and positively associated with accurate health knowledge [36]. Consequently, health literacy may play an important role in preventative screenings because pamphlets and written instructions are often used to explain and justify screening interventions. This is especially significant as CRC is the third leading cause of cancer, which still carries a lot of stigma, and AA/B men are disproportionately impacted [37].

Evidence suggests that health literacy impacts the understanding of health insurance and payment information for AA/B men, which may further contribute to health disparities. A qualitative study by Politi et al. [27] examined the health insurance knowledge and preferences of 51 uninsured adults in urban and rural Missouri, with 69% of participants being Black and 70% making less than USD 15,000. By administering the Rapid Estimate of Adult Literacy in Medicine (REALM)—Short Form, this study found that 47% had marginal or inadequate reading and health literacy levels. Those with greater health literacy demonstrated more accurate health insurance knowledge compared to those with inadequate or marginal health literacy. Health literacy played a role in the understanding of health insurance and billing details. Gender differences have also been observed in the understanding of health insurance information, in that females showed greater understanding of health insurance information. Both health literacy and gender were noted to be risk factors for increased challenges in understanding complex health system and insurance information [27]. This is important, as AA/B men are at heighted risk of chronic health conditions [38,39] and access to health system and insurance information can be critical in managing chronic health conditions.

### 1.3. Digital Health Literacy

The role of digital health literacy or eHealth literacy among AA/B men is important to understand, especially in today’s society. Digital health or eHealth literacy is defined by the World Health Organization (WHO) as the ability to seek, find, understand, and appraise health information from electronic sources and apply the knowledge gained to addressing or solving a health problem [40,41]. Healthcare has become more digitized in recent decades, including increased use of online information and mobile applications [41]. New skills are needed to navigate digital health information, including how to search, assess, choose, and apply health information obtained online [41]. Digital health literacy that focuses on information gathering is referred to as Health 1.0 skills. Digital health literacy that focuses on and interactivity on the web is referred to as Health 2.0 skills [41]. Health 2.0 applications include health communication applications such as e-consultations with healthcare providers, forums to interact with peers, and patient portals to self-monitor health, as well as telehealth platforms to access telehealth appointments [41]. Both Health 1.0 and Health 2.0 skills have the potential to improve health if people have access to technology and can use these skills effectively to obtain relevant health information.

Technology has the potential to either improve healthcare or increase existing health disparities depending upon patients’ abilities to access, use, and understand healthcare technologies [42]. For example, technology may be useful for patients with low health literacy if they are able to engage with the technology. A study by Vollbrecht et al. [42] examined the relationships between access to technology, technology use, and health literacy among hospitalized patients. The study enrolled 502 participants. The mean age was 51, 71% were African Americans, and 52.8% were female. Participants with low health literacy needed more help navigating online tasks. Interestingly, low health literacy was not associated with a lower likelihood of owning a smartphone [42]. This establishes that even patients with low health literacy had access to technology but required additional assistance navigating online tasks [41,42]. These findings suggest that eHealth literacy interventions may be helpful for those with low health literacy and that eHealth interventions should include ways to actively support the navigation of online tasks and building both Health 1.0 and 2.0 skills.

Low levels of eHealth literacy are typically associated with older age, lower socioeconomic status, and a lower education level. Alarmingly, lower levels of eHealth literacy were also associated with being chronically ill. Furthermore, those who had high eHealth literacy had significantly more access to computers and the Internet and were more adept at gaining digital health information, such as employing useful Internet search strategies and evaluating information obtained [43]. Likewise, those with high eHealth literacy demonstrated better understanding of health status, treatment options, and symptoms and were more likely to ask their physicians more questions. These findings highlight how disparities in eHealth literacy and access to technology are influenced by sociodemographic factors and health status, such as being chronically ill, which is particularly relevant for AA/B men who experience higher rates of chronic illness [38,39].

Most importantly, digital health literacy can directly influence health behaviors and health disparities, for better or for worse. Studies have shown that higher levels of digital literacy promoted health status and positive health behaviors such as healthy eating, exercise, and sleep behavior [44]. However, in another study examining literacy disparities related to digital health information, differences in eHealth literacy created new inequities and reinforced existing social differences [43].

### 1.4. Impact of Digital Literacy on AA/B Men’s Health

Digital technology may help to reduce chronic illness and health disparities among AA/B men [45,46,47]. Digital technologies may challenge the limitations of traditional public health and health education programs by tapping into natural social networks among AA/B men [46]. Forces such as racism and medical mistrust may impact health behaviors, such as delaying routine check-ups, blood pressure checks, and cholesterol screenings among AA/B men [48]. In one study, patients who scored higher on health literacy measures were able to find helpful health resources on the Internet, while participants who scored lower could not differentiate between poor-quality and high-quality resources [49]. Self-directed preventative healthcare engagement may serve to protect from medical harm or mistreatment for AA/B men [48] and may prove useful in developing digital health interventions and improving eHealth literacy in AA/B men.

Gender differences were also found in eHealth literacy levels, willingness to participate in research using digital health technology, and engagement with various digital health platforms. For example, women had higher eHealth literacy scores, were more willing to engage in mobile health research, and were more likely to have downloaded a health app then men [49].

These findings suggest that digital technology has the potential to increase access to appropriate healthcare knowledge and that eHealth literacy may serve as a helpful intervention to reduce healthcare disparities among AA/B men. Additionally, these findings point to the need to better understand how AA/B men interact with digital technology and how they interpret the information they receive in order to develop and improve digital literacy interventions that aim to improve AA/B men’s health.

## 2. Materials and Methods

Within the greater medical literature, recognition is growing regarding the value of interventions that improve patient self-management of chronic medical conditions [50,51,52]. CDSMPs are designed to empower individuals with chronic conditions to actively manage their health themselves and improve their quality of life. These programs primarily focus on developing skills and strategies for living day-to-day with a chronic condition, advancing health behaviors, and fostering a sense of self-efficacy [53,54]. CDSMPs are effective in improving health outcomes for individuals with long-term health conditions [53,55,56]. Still, AA/B men—particularly those with low incomes and multiple comorbidities—remain grossly underrepresented in such programs due to systemic barriers and mismatched program designs.

The main goal of this research was to assess opportunities and barriers for developing a culturally and contextually adapted CDSMP to effectively engage low-income AA/B men with multiple chronic conditions who often face intersecting difficulties. These often include limited health literacy, structural racism, and socioeconomic complications. Not only did this study aim at improving knowledge, confidence, and behaviors related to managing chronic conditions, but it also emphasized the crucial impact of low general and health literacy on CDSMPs. This study explored the role of literacy on the participant engagement, program success, and scalability of an adapted CDSMP for a specific population.

### 2.1. Study Design

The overall purpose of this pilot study was to assess the feasibility, acceptability, process, and preliminary outcomes of an adapted CDSMP specifically tailored for low-income AA/B men living with multiple chronic conditions. This pilot study will inform the design of a future large-scale randomized clinical trial (RCT) which can further document knowledge and behavior-based outcomes. The study design was a single-arm pre-/post-test and mixed-methods pilot program. Approximately 45 low-income AA/B men who were living with multiple chronic conditions were initially enrolled in a multi-session, peer-led, adapted CDSMP that aimed to better meet the unique needs of AA/B men.

Because of the limited literature on the efficacy of self-management programs with vulnerable populations, especially those experiencing both poverty (and related issues of low education, unemployment, violence, insecurity, and environmental exposures) and multiple diagnoses of chronic illnesses and/or behavioral health conditions [57], an engaged scholarship approach was employed. We partnered with local community health workers, non-profit agencies serving AA/B men, and AA/B men themselves on all aspects of this pilot study. Additionally, the intervention was led by formally trained peer facilitators who also identified as AA/B men living with multiple chronic conditions. These peer facilitators also provided regular feedback and insight into adapting the curriculum.

This pilot intervention entitled “Healthy Together” utilized the foundational concepts of the CDSMP developed by Lorig et al. at the Stanford Patient Education Center [54,55,56,58]. “Healthy Together” focuses on a wide range of chronic conditions, such as high blood pressure, hyperlipidemia, HIV, diabetes, COPD, and arthritis, as well as mental health conditions [59]. Despite the diversity in diagnoses, “Healthy Together” includes specific self-management tasks that have been found to be common across chronic disease conditions [55,56,60,61,62]. The key elements of the intervention consisted of regular action planning and feedback, modeling of behaviors and problem-solving by participants, reinterpretation of symptoms, and training in specific disease management techniques. Data were collected through surveys prior to the intervention, immediately post-intervention, and at three months follow-up, along with one-on-one semi-structured interviews at the conclusion of the program with both the participants and peer facilitators.

### 2.2. Setting and Participants

The sample was recruited through convenience sampling in waiting rooms and via flyers posted in outpatient clinics and other health facilities serving AA/B men. Email scripts describing the study, its eligibility criteria, and its voluntary nature were also utilized to reach out to database lists and contacts. This dual approach has been found optimal for recruiting vulnerable populations for health behavior interventions [63]. The inclusion criteria encompassed being over the age of 18; meeting the specified criteria for low-income status (e.g., eligible for Medicaid or uninsured and living below 200% of the federal poverty line); self-identifying as an AA/B man; self-reporting at least two chronic conditions (either physical or mental); and having the capacity to provide informed consent. Because gaining process feedback during this pilot phase was critical, inclusion criteria were kept broad to maximize participation and optimize generalizability. After a brief eligibility interview conducted by project staff that also outlined the purpose, expectations, and incentives (which included a monetary stipend for each workshop attended; free healthy snacks, lunch, or dinner at each workshop; and transportation reimbursement), participants consented into the study.

Additionally, since this was a simple pilot study with a single-arm pre-/post-test design to primarily assess feasibility (e.g., recruitment, retention, and acceptability), estimate parameters (e.g., standard deviations and effect sizes) for future studies, identify and resolve logistical issues, and refine intervention protocols and data collection methods, and not a hypothesis-testing study, formal power calculations were not required. Instead, for a medium effect size (Cohen’s d = 0.5) with an 80% power at a 0.05 confidence level, at least 34 participants were needed. We also added an additional 15–20% to account for the attrition, bringing the appropriate sample size to a minimum of 40 participants [64]. Previous research has suggested that pilot studies ideally contain 30–50 participants, and a sample group of around 12 per workshop session is typically recommended for interventions such as this to ensure that the data are informative and justify a larger study [65].

Ultimately, “Healthy Together” was implemented face-to-face at local community organizations that served AA/B men with a total of 42 participants completing one of four separate workshop series offered between 2020 and 2022. Each workshop series accommodated about 10–15 participants. Each “Healthy Together” workshop series consisted of three extended group sessions. These sessions were often scheduled around other organizational events to increase attendance and participation, with additional one-on-one coaching sessions conducted by the peer facilitators in between sessions. Each group session lasted about four hours and met each week for three consecutive weeks to accommodate availability. The follow-up one-on-one health coaching sessions with one of the peer facilitators lasted 30–60 min each and occurred face-to-face or virtually during the session weeks.

Each group session was led by two trained “Healthy Together” peer facilitators who both led the group workshops together and provided one-on-one health coaching sessions individually. Additionally, supplemental written (e.g., handouts, resource guides, and workbooks), visual (e.g., posters), and electronic (e.g., personal health records, websites, and apps like Fooducate, MyFitnessPal, and HealthVault) materials were developed and curated, further supporting the content shared in the group workshop sessions and the one-on-one health coaching sessions.

### 2.3. Program Adaptation Process

While the core structure of general self-management was maintained, several key modifications were adapted to the unique needs and challenges of low-income AA/B men with multiple morbidities informed by preliminary data gathering. Because of the potential gaps in health literacy and *cognitive limitations* [66,67], the curriculum was simplified, supporting low reading levels and cultural differences, particularly regarding disease-specific information, medications, managing medical appointments, navigating the healthcare system, preventative healthcare, communication, dietary intake, and physical activities. To improve motivation and engagement in care leading to stronger *self-efficacy* and *coping* skills, additional supportive techniques were utilized, such as health coaching, goal setting, problem-solving, family involvement, social media, technology, journaling/storytelling, and personal health records. Thus, *social support* opportunities increased. Specific culturally appropriate materials were added, including emphasizing the connection between physical and mental health, the importance of coordinating information about multiple medical conditions across varied providers as well as family and friends, and informed decision-making and ethical dilemmas. The diet and exercise sections were significantly modified to address the high rates of poverty and social disadvantage in this population. These sections provided realistic strategies for purchasing healthy food on a budget (e.g., using food stamps), healthy cooking in the midst of housing and/or food insecurity, and allowing participants to safely exercise in their own homes and neighborhoods.

Utilizing peer leaders for group facilitation as well as one-on-one coaching and support was also a major cornerstone of this intervention. For this, we borrowed aspects of the certified peer specialist (CPS) program used in the mental health consumer workforce to promote mental health recovery and well-being [68,69]. Amidst growing concern in the mental health community about elevated morbidity and premature mortality [70], mental health consumer leaders are increasingly calling for efforts for the integration of mental health, physical health, and wellness into existing consumer recovery programs, many of which are highly populated with AA/B men [71]. However, no evidence-based interventions are currently available for this.

Due to COVID-19, we also had to make some logical adaptations which were found to be beneficial, both short- and long-term. Due to scheduling challenges and the need for COVID-19 transmission protections, we truncated the intervention structure from one two-hour workshop per week for six weeks to three four-hour workshops per week for three weeks. For more hands-on learning amidst social distancing, we also included a separate cooking demonstration class and a group exercise class for each workshop cohort. Also, to take advantage of the public health lessons from COVID-19, we also incorporated sections on infectious disease transmission, basic hygiene, immunizations, medical research, media reporting on health crises, and communicating concerns and difficult issues with healthcare professionals—all of which are not usually included in standard CDSMPs.

### 2.4. Data Collection

Since this pilot intervention focused mostly on process assessment and feasibility, the data collection employed a mixed-methods approach. Quantitative survey data were collected from 42 participants via a pre-test survey before the intervention started. Once the intervention was completed and at a three-month follow-up, post-test surveys were collected again. The survey included questions about basic demographics, health knowledge, health behaviors, health status, healthcare utilization, self-efficacy and confidence, and patient activation and engagement. Simple “health fair-type” biometrics were also collected by nursing students, including height, weight, BMI, blood pressure, pulse rate, blood cholesterol, glucose levels, and health risk identification. This information was shared with the participants as a baseline to discuss with their healthcare providers.

Qualitative measures included 25–60–90 min one-on-one interviews between participants and peer facilitators at the conclusion of the intervention. The semi-structured interviews included questions about relevance and cultural fit, accessibility and literacy, the peer-led format, behavior change and impact, barriers and challenges, motivation and engagement, trust and safety, and suggestions for improvement. Again, interviews were scheduled based on convenience and participant availability, particularly during the height of the COVID-19 pandemic. These data were digitally recorded and transcribed to document participant experiences, barriers, and suggested improvements.

All “Healthy Together” sessions were digitally recorded and observed by project staff. Process-specific measures such as session attendance, satisfaction, comfort level, and general understanding of the material were also collected throughout the intervention. Peer facilitators also took detailed notes regarding their health coaching and one-on-one interactions. These data were further assessed to ensure intervention/curriculum adherence, fidelity, and quality assurance.

### 2.5. Data Analysis

Raw survey data were carefully reviewed for completeness and then transferred to SPSS 29. Because this was a small, controlled study with low overall literacy rates, project staff were directly involved with the collection of survey data. To minimize missing data upfront, preventative strategies were used, such as simplifying data collection using interviewer-administered surveys, sending out reminders and follow-ups during sessions, offering incentives, and building rapport and trust with the participants.

Descriptive statistics, including demographics, baseline characteristics, and feasibility metrics were calculated. Composite scores for changes from pre-test to post-test and t-tests were noted, both at the end of the intervention and at the three-month follow-up, for each measure. This helped define the relationship between “Healthy Together” and improved knowledge, healthy behaviors, health status, self-efficacy, healthcare utilization, and patient activation.

Qualitative data through interviews with “Healthy Together” participants and peer leaders were recorded, transcribed, and inputted into NVivo 14. A modified grounded theory methods (GTM) approach generated themes grounded in participant experience while also incorporating structures from existing theories, including health literacy, self-efficacy, and cultural tailoring. Modified GTM allows for themes to emerge inductively from the data while also drawing on the relevant pre-existing theoretical frameworks. Initial open coding, memo writing, axial coding, selective coding, and theme development continued until theoretical saturation was achieved. This analysis ultimately supported additional code frequency counts, relationships among codes, and further data visualizations.

With modified GTM, reliability refers to the consistency, transparency, and dependability of the coding and analytic process [72]. To ensure reliability for the qualitative data, at least two independent researchers participated in the coding. Each one independently coded the data, then they met to compare and reconcile any differences. Intercoder agreement checks were performed, and disagreements were documented and resolved. Likewise, the trustworthiness of the data was further promoted through incorporating peer debriefings and an audit trail to track decisions and changes over time. Emergent themes were reviewed by the research team in relation to the original research questions about feasibility and potential opportunities and barriers, as well as the overarching behavior-change theoretical frameworks, to ensure coherence and relevance. Data were also triangulated, and initial findings were shared with key stakeholders for additional checking and validation of interpretations, further enhancing credibility and contextual relevance.

Although this was only a pilot study, the reliability and validity of the data were further addressed through consistent implementation. Program fidelity was ensured through checklists, facilitator observations, and session recordings. Data collection tools were pre-tested, especially considering low literacy levels. The same procedures, locations, and conditions were utilized for data collection to reduce variability, and project staff were trained in effective data collection to minimize literacy barriers and bias. Patterns across data types were continuously compared and integrated into program revisions.

## 3. Results

Regarding the general characteristics of the 42 participants who completed the entire “Healthy Together” program, the mean age was 53 (ages ranged from 22 to 80). Common self-reported comorbidities included hypertension (93%), high cholesterol (81%), type 2 diabetes (53%), arthritis (48%), depression (40%), and HIV (17%). Most participants reported being single, unemployed, and experiencing food and/or housing insecurity. Moreover, 71% reported no regular primary care provider, and 64% reported reading difficulties or discomfort with written materials. All the participants indicated that they felt comfortable using technology and had reliable access to devices (e.g., smartphones, tablets, and/or computers) through their own personal ownership, family and friends, or the local public library.

Although the quantitative measures assessing self-efficacy, medication adherence, physical activity levels, nutrition, symptom severity, and health-related quality of life showed modest improvements, none proved statistically significant. However, this paper does not focus on actual health outcomes and impact at this initial stage. Instead, the data presented here generate preliminary insights that will inform the design of a future full-scale intervention trial. From a satisfaction perspective, nearly all the participants indicated that they enjoyed participating in the workshops and would recommend participating in “Healthy Together” to a friend. Due to thorough preventative measures, consistent reminders and support by peer leaders, and incentives, attrition was not a problem in this case, with 93% of participants completing all aspects of the program. In fact, high levels of overall satisfaction and participation led to the scheduling of several additional informal “reunion” meetings after the intervention so that they could continue to meet, engage in healthier behaviors together, and support each other.

One crucial and overarching issue that emerged consistently was how low health literacy served as a significant barrier to CDSMPs. Without proper adaptations specifically addressing literacy challenges, participants expressed difficulty in understanding key concepts related to successful self-management. In turn, their desire to fully participate in “Healthy Together” was negatively impacted. Throughout the intervention, topics had to be verbally reframed or simplified to support comprehension. This highlighted a reliance on oral learning styles and peer support over traditional text-based formats. Participants clearly required additional assistance to avoid confusion or embarrassment during both the intervention and data collection components.

More specifically, major literacy-related themes emerging from the data included the following: (1) Visuals and oral storytelling enhanced perceived understanding. (2) Participants appreciated oral delivery and interactive formats over traditional didactic learning, which was seen to be boring and ineffective. (3) Participants felt that the program and the scenarios were timely (especially in the midst of the COVID-19 pandemic), relatable, and culturally tailored. (4) Facilitators and peers with similar lived experiences were crucial to enhancing relevance. (5) Many participants did not readily disclose their literacy limitations before and expressed feeling embarrassed and/or stigmatized. (6) Participants expressed easy access to digital devices and technologies but often required help assessing and navigating information sources and other resources. (7) Camaraderie and shared experiences created a strong group bond, further encouraging active engagement and sustainability.

## 4. Limitations

Since this was an initial pilot study to assess the overall process and feasibility, the small sample size limited the generalizability of the findings, particularly for other geographic regions or demographics. This study was likely influenced by selection bias, in that participation in “Healthy Together” was voluntary and recruitment relied on referrals, community partners, and clinic outreach. The individuals who choose to enroll may differ systematically from those who did not. This study also occurred during the height of COVID-19, forcing repeated recalibration of recruitment plans and timelines to ensure the integrity of the intervention and research. Although a three-month follow-up was conducted, long-term impacts on chronic disease self-management behaviors, health outcomes, and healthcare utilization were not fully assessed, leaving questions about the sustainability of behavior change. Data on health behaviors and outcomes relied heavily on self-reported data, which are susceptible to recall bias and social desirability bias—particularly in a group setting where participants may have wanted to demonstrate healthy choices from the beginning—making changes in outcomes and impact more difficult to ascertain.

Although low and zero literacy levels were clearly identified as significant challenges, participants were not formally assessed using standardized health literacy or reading comprehension tools, resulting in limitations in terms of further tailoring and evaluation. Literacy-related adaptations were developed reactively and differed slightly across groups depending on facilitator skills and available resources. Likewise, additional technology and digital tools were introduced mid-stream as a reaction to observed behaviors and the complications set forward by COVID-19, so their effectiveness as standalone components was not determined or formally evaluated. The impact of the facilitators (e.g., communication style, cultural concordance, etc.) may have also contributed to outcomes, but this was not systematically evaluated. Group-based programs are also commonly influenced by dominant voices or peer pressure, possibility skewing engagement or self-reported outcomes—especially in a setting where stigma around literacy and health exists.

The initial findings from this pilot/feasibility study will be utilized to develop a more refined and robust program that will be formally tested using a randomized clinical trial design with control or comparison groups. Consequently, detailed outcomes and other external factors, such as concurrent medical care, social support, or other influences that may have played a role in change, will be assessed.

## 5. Discussion

The central aim of this pilot study was to better understand how existing CDSMP curricula could be adapted to be more culturally relevant, gender-sensitive, and accessible for a historically underserved population. One major finding was that this pilot study demonstrated that literacy challenges can be a hidden but critical barrier to chronic disease self-management among underserved AA/B men. The unanticipated need to actively and thoroughly address literacy—basic literacy, health literacy, and digital literacy—illuminated broader issues of health equity and the importance of a more user-centered design.

### 5.1. Enhancing Literacy in CDSMPs

This study elucidated that low literacy (and sometimes no literacy) was more prevalent than anticipated. Moreover, existing written materials, toolkits, and templates for traditional CDSMPs alone were not sufficient [55,56]. Facilitators and other peers played a critical translational role; shame and stigma around literacy affected participation, especially if left unaddressed; literacy at all levels must be explicitly integrated into program design; and literacy rates likely impact program outcomes and data quality. Ensuring appropriate reading levels was simply not enough. Carefully assessing literacy early on, preferably utilizing formal assessment tools like REALM (Rapid Estimate of Adult Literacy in Medicine), TOFHLA (Test of Functional Health Literacy in Adults), eHeals (eHealth Literacy Scale), or the Health Literacy Questionnaire (HLQ), is imperative. Significant assumptions about literacy may result in further exclusion and embarrassment. Interventions like this one should be designed for oral/visual learning using multimedia, storytelling, oral reinforcement, and interviewer-implemented data collection. In addition to the curriculum, peer facilitators should be trained in inclusive communication to enhance interventions, build trust, encourage active participation, and reduce stigma. Community voices should be engaged throughout studies to ensure that adaptations are effective, relevant, and respectful. Most importantly, cultural and contextual relevance matters: participants connected better with others who shared a similar background and lived experiences.

### 5.2. Specific Health Literacy Strategies for AA/B Men

It has been noted that culturally sensitive approaches are needed in applying interventions with AA/B men and that self-management and peer-led approaches may be effective methods to help overcome the barrier of medical mistrust [73]. Non-traditional settings outside of healthcare facilities may be useful in recruiting for and implementing health interventions for low-income AA/B men [74]. Additionally, oral traditions and other methods of communication may serve as powerful means of cultural communication. Overall, health education/behavior change interventions like CDSMPs for low-income AA/B men should stress self-efficacy, peer leadership, oral communication, active learning, various teaching and learning strategies, and non-traditional settings to communicate and deliver health information [46,48,74].

As mentioned earlier, inequities in health literacy exist for AA/B men, and research shows that medical professionals are not fully aware of these disparities. Strategies tailored to enhance health literacy among AA/B men may include educating medical professionals about low literacy and how to effectively mitigate this challenge [29]. Similarly, suggestions to improve health literacy include increasing healthcare providers’ awareness of problems patients may have with using digital technology to access and understand health information [41]. Healthcare professionals can be encouraged to provide a list of credible websites and other resources that contain helpful information [49].

### 5.3. Improving Digital Health Literacy

Previous research findings, as well as this pilot study, show that high numbers of low-income AA/B men have access to digital technologies such as smartphones, tablets, and/or computers [49]. Low literacy is not associated with a lower likelihood of owning a device [42]. These findings suggest that interventions for AA/B men with low health literacy should improve eHealth literacy. Interventions that highlight digital health literacy offer promising strategies to improve the overall health and well-being of AA/B men. To apply digital health literacy interventions effectively to low-income AA/B men and other marginalized populations, attention needs to be given to ensuring that the accessible and culturally appropriate digital health platforms and technologies are used to deliver healthcare information. Additionally, it is important for researchers and developers of health interventions to deliver these interventions via platforms that are utilized and deemed trustworthy by AA/B men and members of marginalized communities, as well as those who care for them [75].

Differences in digital health literacy should be further identified and incorporated in health intervention planning. Studies have shown that those with low eHealth literacy demonstrated higher perceived trust in online channels such as YouTube and Twitter, while others with high eHealth demonstrated higher perceived trust in online government, educational, and religious intuitions [48]. Older adults are often less trustworthy regarding online sources, so health providers and researchers should work to train older adults in how to assess the creditability of online health resources [46]. Gender differences in digital health literary are typically not found overall; however, when individuals with lower eHealth literacy were compared, women demonstrated more trust in online health information [75]. Higher education and more engagement with online platforms (e.g., Facebook, twitter, Pinterest, blogs, support groups, and YouTube) was associated with greater perceived trust in online health communication. When controlling socioeconomic status and social media usage, higher eHealth literacy predicted greater perceived trust [46,48,75].

These findings speak to the need to understand the diverse experiences of AA/B men in terms of engagement with digital technology, eHealth literacy, and perceived trust in various digital technology platforms. By increasing the understanding of which platforms are more trustworthy among low-income AA/B men with varying eHealth literacy levels, researchers, health professionals, and health systems will be better able to develop digital technology health communication interventions that match the eHealth literacy level, socio-demographic factors, and perceived trust in various digital platforms of AA/B men [75].

Finally, suggestions to improve eHealth literacy include improving digital design to better meet consumer needs [43] and improving usability in web design on digital health platforms [41]. It is also important that digital content be culturally appropriate for low-income AA/B men; therefore, including end users with varying health literacy and eHealth literacy levels in digital literacy platform development and design is essential.

### 5.4. Digital Health Literacy Post-COVID-19

As seen in this small pilot study, the COVID-19 pandemic significantly transformed the landscape of health education by accelerating the adoption of digital technologies. Traditional in-person interventions were rapidly disrupted due to public health restrictions, prompting a widespread pivot to virtual or hybrid platforms for health promotion and education. This shift catalyzed innovation in the use of telehealth, mobile applications, online workshops, and social media to disseminate health information and engage communities.

Digital methods provided critical continuity for health education during periods of lockdown and social distancing, offering flexible, scalable, and often cost-effective alternatives. Interventions targeting chronic disease management, mental health, and preventive care increasingly leveraged digital platforms to better reach vulnerable populations, including those in rural and underserved areas [76,77].

However, the shift also revealed persistent digital divides. Barriers such as limited Internet access, low digital literacy, and socioeconomic disparities disproportionately affected historically marginalized populations, raising concerns about equity in digital health education [78]. Additionally, while digital interventions proved effective in many contexts, challenges in maintaining engagement, ensuring cultural relevance, and evaluating outcomes in real time remained significant [79]. Overall, the pandemic served as both a catalyst and a stress test for digital health education, highlighting the potential of technology-enabled interventions while underscoring the need for inclusive, accessible, and sustainable digital health strategies.

Beyond digital health literacy changes due to COVID-19, it is critical to note that general literacy was also significantly impacted, which is likely to continue in years to come. COVID-19 has led to significant declines in literacy rates globally. In the United States, the National Assessment of Educational Progress (NAEP) reported that reading scores dropped between 2020 and 2022, marking the largest decline since 1990 [80]. Likewise, the reading proficiency levels for the fourth and eighth grades fell to 31% and 30%, respectively, the lowest levels since 1994 [80]. These setbacks are attributed to prolonged school closures, limited access to remote learning, and increased absenteeism, disproportionately affecting disadvantaged communities. Experts warn that without targeted interventions these literacy losses could have long-term implications for health equity. As a result, it is critical to utilize this new information to plan future health interventions.

### 5.5. Lingering Issues

Chronic diseases disproportionately impact low-income AA/B men, who face some of the highest rates of morbidity and premature mortality in the United States. Compared to other racial and gender groups, AA/B men are more likely to be diagnosed with multiple chronic conditions such as hypertension, diabetes, and cardiovascular disease, often at earlier ages and with worse outcomes [81,82]. These disparities are compounded by structural factors, including poverty, limited access to healthcare, systemic racism, and medical mistrust, which collectively undermine chronic disease management and patient engagement [81,83].

CDSMPs, originally developed by Lorig et al., have been shown to improve self-efficacy, reduce healthcare utilization, and enhance quality of life for individuals with chronic conditions [55,84]. Nevertheless, the existing literature suggests that CDSMPs have not been adequately tailored to or evaluated for low-income AA/B men, whose unique sociocultural, economic, and health system barriers require targeted adaptations. Most published CDSMP studies involve general populations or disproportionately include white, female, and higher-income participants [85,86], making it difficult to assess program effectiveness and feasibility among marginalized men.

Furthermore, a preliminary scan of the literature indicates inconsistent reporting on recruitment strategies, cultural tailoring, literacy accommodations, and long-term engagement in chronic disease self-management, especially among low-income AA/B men. Without a rigorous and focused systematic review, the field lacks a comprehensive synthesis of what has been tried, what has worked, and where critical gaps remain. A systematic review is therefore urgently needed to (1) identify and evaluate the existing CDSMP interventions that specifically target or include low-income AA/B men; (2) assess the extent to which programs are culturally and contextually adapted for this population; (3) determine outcomes related to feasibility and acceptability, in addition to health improvement; (4) highlight methodological strengths and limitations in the literature; and (5) guide the design of future interventions and inform funding, policy, and practice aimed at reducing chronic disease disparities through basic literacy, health literacy, and digital literacy. Such a review will help move the field beyond a one-size-fits-all approach and toward precision public health models that account for gender, race, income, and lived experience in chronic disease management.

## 6. Conclusions

Adapting a CDSMP for low-income AA/B men with multiple chronic conditions revealed an urgent need to address major literacy barriers, spanning basic literacy, health literacy, and digital literacy. Future interventions must carefully consider literacy at all levels as well as cultural relevance, community-informed design, and creative delivery strategies from the start. Structural barriers such as systemic racism, limited access to quality education, and socioeconomic disparities continue to impact overall health communication and engagement. When conducted properly, CDSMPs and other similar health interventions can empower marginalized communities and improve health outcomes by offering person-centered, accessible, and interactive health information. Investing in historically marginalized communities, like low-income AA/B men, by providing uniquely tailored programs containing accurate and effective knowledge, tools, and support to critically engage with health information is not only a matter of individual well-being but a vital step towards achieving broader health equity.

## Data Availability

The original contributions presented in this study are included in the article. Further inquiries can be directed to the corresponding author.

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
