# Peer review of "Black Men and Health Literacy: Strategies for Improvement in a Digital Age Through the Adaptation of a Chronic Disease Self-Management Program"

_ijerph, 2025, doi:10.3390/ijerph22071153_

Round 1

Reviewer 1 Report

Comments and Suggestions for Authors

General Assessment

The manuscript addresses a relevant public health issue of health literacy and strategies for improving men’s health. The authors aim to explore the relationship between social determinants and health outcomes among a marginalized group, a topic of continuing importance. In general the study design is acceptable and the topic suitable for the journal but several areas require clarification, improvement, or short down.

Clarity of Aim and Research Question

The aim of the study needs to be more clearly stated. With a more concise and specific framing of the research objectives.

Introduction

Generally, shorten down the text in the introduction. The description of studies should only give an overview of findings.

Methods

A more defined and detailed description of the study is needed to give an understanding of how the study was conducted.

How were the population were recruited. The inclusion criteria and exclusion rationale should be reported. Where were the study conducted – geographical area. Who did the recruitment.

Methods for collecting data like self-reported questionnaires and interview guides are not described clearly. Also the description of key variables is very general.

Statistical methods must be described thoroughly. This includes detailed information about variables used in models and covariate adjustment. Explanation of how missing data were handled. Who did the analysis – explain key stakeholders.

Results

The interpretation of the results only highlights key findings. This gives an unclear picture of the results. Please give a more  detailed description of the findings from the pilot study.

Discussion

The discussion of general strategies for improving health literacy for the study group are satisfactory, but I miss a discussion of the results from the study except from the summary in the lessons learned section.

Comments on the Quality of English Language

Language and Style

Several grammatical and syntactical issues were observed. Thorough language editing is recommended to improve clarity and fluency.

Note text in line: 36, 63, 240

Author Response

RESPONSES TO REVIEWER COMMENTS

General Assessment

The manuscript addresses a relevant public health issue of health literacy and strategies for improving men’s health. The authors aim to explore the relationship between social determinants and health outcomes among a marginalized group, a topic of continuing importance. In general the study design is acceptable and the topic suitable for the journal but several areas require clarification, improvement, or short down.

Clarity of Aim and Research Question

The aim of the study needs to be more clearly stated. With a more concise and specific framing of the research objectives.

Response: This was addressed about being more clear about the relationship between adapting a new CDSMP for low-income African American/Black men and literacy.  The overall goal of this pilot study was feasibility. 

Introduction

Generally, shorten down the text in the introduction. The description of studies should only give an overview of findings.

Response: Introduction was shorten by more than 2 pages and relevant findings were better highlighted. 

Methods

A more defined and detailed description of the study is needed to give an understanding of how the study was conducted.

Response: This was addressed throughout the materials and methods section, focusing more on the pilot components of this project. 

How were the population were recruited. The inclusion criteria and exclusion rationale should be reported. Where were the study conducted – geographical area. Who did the recruitment.

Response: This was addressed in the setting and participants section by adding in this information on recruitment, inclusion and exclusion criteria, geographical area, and who did the recruitment. 

Methods for collecting data like self-reported questionnaires and interview guides are not described clearly. Also the description of key variables is very general.

Response: More information was provided about this in the data collection section. 

Statistical methods must be described thoroughly. This includes detailed information about variables used in models and covariate adjustment. Explanation of how missing data were handled. Who did the analysis – explain key stakeholders.

Response: Although this study was a pilot study to assess feasibility and not outcomes, we did include more details about the statistical methods, missing data, and data analysis. 

Results

The interpretation of the results only highlights key findings. This gives an unclear picture of the results. Please give a more  detailed description of the findings from the pilot study.

Response: We provided more emphasis on the results that informed the feasibility study and will be used for a larger RCT study in the future. 

Discussion

The discussion of general strategies for improving health literacy for the study group are satisfactory, but I miss a discussion of the results from the study except from the summary in the lessons learned section.

Response: We emphasized the overall results that supported the feasibility goal of this study, especially those linked the adapting the CDSMP and literacy. 

Comments on the Quality of English Language

Language and Style

Several grammatical and syntactical issues were observed. Thorough language editing is recommended to improve clarity and fluency.

Response: We corrected this throughout the paper. 

Reviewer 2 Report

Comments and Suggestions for Authors

This manuscript addresses an important and timely topic: improving health and digital health literacy among African American/Black men, a population historically underserved and disproportionately affected by chronic disease. The authors present a mixed-methods feasibility study of an adapted Chronic Disease Self-Management Program (CDSMP) tailored for this demographic.

The study is clearly structured and well-written, with rich context, relevant literature, and community-engaged methodology. However, while the article is commendable in many ways, several methodological, analytical, and presentation issues require revision or clarification before it can be considered for publication. There are instances of typographic errors, repetition (e.g., "African American/Black men" is overused without abbreviation), and overuse of parentheses.

While the topic is highly relevant, the paper does not always clearly distinguish what this study contributes versus what prior literature has established. Some results reiterate well-known patterns (e.g., low literacy, systemic racism) without clearly showing how the intervention added value.

The title suggests a broad exploration of “strategies,” but the actual intervention is narrowly scoped (42 participants, limited settings). Consider adjusting the title to better reflect the scope.

  1. Abstract and Introduction

The background section is quite long and heavily cited but lacks critical synthesis. It reads more like a literature review than an introduction leading into a specific study. Consider shortening and better funnelling toward the research gap.

The introduction would benefit from a more explicit statement of research objectives and hypotheses. Currently, the study purpose is buried in the middle of the paper.

  1. Methodology

The study design is described as a “single arm pre/post-test feasibility pilot study,” but key methodological details are vague:

What were the specific quantitative instruments used? How were validity and reliability ensured?

What specific qualitative approach or theoretical framework guided the interviews (e.g., grounded theory, thematic analysis)?

Participant inclusion/exclusion criteria could be clarified.

Statistical power and sample size justification are missing, even for a feasibility study.

The adaptation of the CDSMP is well described, but the process for evaluating those adaptations is insufficiently explained. Were adaptations tracked across cohorts?

  1. Results

Quantitative results are underreported and underanalyzed. Authors state “none proved statistically significant” but provide no test statistics, confidence intervals, or effect sizes. Without these, it is difficult to assess the program’s impact.

Consider including a table summarizing pre- and post-intervention quantitative measures with descriptive and inferential statistics.

Qualitative results are compelling, but analytic rigor needs clarification:

How were codes developed?

How many coders?

Were member checks performed?

Was saturation achieved?

  1. Discussion

The discussion lacks sufficient critical reflection on the null quantitative results. What does this say about intervention effectiveness?

More direct linkage between findings and the intervention's components (e.g., storytelling, visual aids, peer support) would strengthen the interpretation.

Several assertions (e.g., about eHealth trust patterns) feel more speculative or based on secondary literature rather than grounded in this study’s findings.

  1. Limitations

The limitations section would benefit from a discussion of potential selection bias. Participants were recruited from clinics and may already be more health-engaged than the general population.

Impact of the peer facilitators is mentioned but not examined—this is a missed opportunity.

  1. Conclusion and Implications

Policy and practice implications are somewhat general. The authors could propose more specific next steps for scaling up or testing the adapted CDSMP (e.g., in an RCT, with tech enhancements).

Author Response

RESPONSES TO REVIEWER COMMENTS

This manuscript addresses an important and timely topic: improving health and digital health literacy among African American/Black men, a population historically underserved and disproportionately affected by chronic disease. The authors present a mixed-methods feasibility study of an adapted Chronic Disease Self-Management Program (CDSMP) tailored for this demographic.

The study is clearly structured and well-written, with rich context, relevant literature, and community-engaged methodology. However, while the article is commendable in many ways, several methodological, analytical, and presentation issues require revision or clarification before it can be considered for publication. There are instances of typographic errors, repetition (e.g., "African American/Black men" is overused without abbreviation), and overuse of parentheses.

Response: We added abbreviation  for AA/B and scaled down the use of parentheses throughout the paper. 

 While the topic is highly relevant, the paper does not always clearly distinguish what this study contributes versus what prior literature has established. Some results reiterate well-known patterns (e.g., low literacy, systemic racism) without clearly showing how the intervention added value.

Response: CDSMP adaptations have not included specific demographics such as race/ethnicity, gender, and socioeconomic status.  CDSMPs usually take on a “one size fits all” approach in that those living with chronic conditions are more similar.  We emphasized this throughout the paper that this informs more person-centered interventions. 

The title suggests a broad exploration of “strategies,” but the actual intervention is narrowly scoped (42 participants, limited settings). Consider adjusting the title to better reflect the scope.

Response: We adjusted the title to include the scope of adapting a CDSMP

  1. Abstract and Introduction

The background section is quite long and heavily cited but lacks critical synthesis. It reads more like a literature review than an introduction leading into a specific study. Consider shortening and better funnelling toward the research gap.

The introduction would benefit from a more explicit statement of research objectives and hypotheses. Currently, the study purpose is buried in the middle of the paper.

Response: We shorted the background/introduction section eliminating more than 2 pages and focused more on the critical analysis and synthesis than the studies themselves.  We also highlighted the research goals and objectives for a pilot feasibility study. 

  1. Methodology

The study design is described as a “single arm pre/post-test feasibility pilot study,” but key methodological details are vague:

What were the specific quantitative instruments used? How were validity and reliability ensured?

Response: We added details about this in the data collection section and how validity and reliability were ensured in the data analysis section. 

What specific qualitative approach or theoretical framework guided the interviews (e.g., grounded theory, thematic analysis)?

Response: We added more information about the modified GTM approach that was utilized in the qualitative analysis component. 

Participant inclusion/exclusion criteria could be clarified.

Response: Inclusion/exclusion criteria were specified in the setting and participants section. 

Statistical power and sample size justification are missing, even for a feasibility study.

Response: This was explained in more detail (power and sample size justification) in the setting and participants section. 

The adaptation of the CDSMP is well described, but the process for evaluating those adaptations is insufficiently explained. Were adaptations tracked across cohorts?

Response: This was explained more as we discussed fidelity in the data analysis section. 

  1. Results

Quantitative results are underreported and underanalyzed. Authors state “none proved statistically significant” but provide no test statistics, confidence intervals, or effect sizes. Without these, it is difficult to assess the program’s impact.

Response: The purpose of this study was a pilot and feasibility study, not to document outcomes and impact (this will be addressed in future papers).   We better clarified the overall scope of this project and results for a specific pilot feasibility study. 

Consider including a table summarizing pre- and post-intervention quantitative measures with descriptive and inferential statistics.

Response: Given this was a pilot feasibility study that found the importance of literacy when adapting a CDSMP for low-income African American/Black men, we don’t want to conflate the study by including data on outcomes and impact (this will be addressed more thoroughly in future papers).

Qualitative results are compelling, but analytic rigor needs clarification:

How were codes developed?

How many coders?

Were member checks performed?

Was saturation achieved?

Response: Codes, coders, checks and saturation were all further clarified in the data analysis discussion on the qualitative measures. 

  1. Discussion

The discussion lacks sufficient critical reflection on the null quantitative results. What does this say about intervention effectiveness?

Response: Since this was more of a process/pilot/feasibility study, we focused this paper only on these measures.  Outcomes will be addressed in future papers.  Since behavior change measures were not found to be significantly significant, we focused more on satisfaction and engagement of the participants and how this will inform future iterations and adaptations for this curriculum. 

More direct linkage between findings and the intervention's components (e.g., storytelling, visual aids, peer support) would strengthen the interpretation.

Response: Further clarity on the specific adaptations was included.

Several assertions (e.g., about eHealth trust patterns) feel more speculative or based on secondary literature rather than grounded in this study’s findings.

Response: We added more support on these assertions grounded in our data and analysis. 

  1. Limitations

The limitations section would benefit from a discussion of potential selection bias. Participants were recruited from clinics and may already be more health-engaged than the general population.

Impact of the peer facilitators is mentioned but not examined—this is a missed opportunity.

Response: A discussion on selection bias and the impact of the peer facilitators were added to limitations. 

  1. Conclusion and Implications

Policy and practice implications are somewhat general. The authors could propose more specific next steps for scaling up or testing the adapted CDSMP (e.g., in an RCT, with tech enhancements).

Response: More information about next steps, including an RCT, were added here. 

Round 2

Reviewer 1 Report

Comments and Suggestions for Authors

No more comments.